# Subgenomic RNA profiling suggests novel mechanism in coronavirus gene regulation and host adaption

Lin Lyu[1],*, Ru Feng[1],*, Mingnan Zhang[1],*, Xiaoqing Xie[1], Yinjing Liao[2], Yanjiao Zhou[3], Xiaokui Guo[1], Bing Su[1], Yair Dorsett[3] ⓘ, Lei Chen[1] ⓘ

**Fundamental to viral biology is identification and annotation of viral genes and their function. Determining the level of coronavirus gene expression is inherently difficult due to the positive stranded RNA genome and the identification of subgenomic RNAs (sgRNAs) that are required for expression of most viral genes. We developed a bioinformatic pipeline to analyze metatranscriptomic data from 20 independent studies encompassing 588 individual samples and 10 coronavirus species. This comparative analysis defined a core sgRNA repertoire for SARS-CoV-2 and found novel sgRNAs that could encode functional short peptides. Relevant to coronavirus infectivity and transmission, we also observed that the ratio of Spike sgRNA to Nucleocapsid one is highest in SARS-CoV-2, among the β-coronaviruses examined. Furthermore, the adjustment of this ratio can be made by modifications to the viral RNA replication machinery, representing a form of viral gene regulation that may be involved in host adaption.**

## Introduction

Coronavirus disease 2019 (COVID-19) reached pandemic levels beginning March 2020 and brought unprecedented devastation to human lives and the global economy (WHO, 2020). The causative agent is severe acute respiratory syndrome corona virus-2 (SARS-COV-2), a β coronavirus similar to Middle East respiratory syndrome coronavirus (MERS-CoV), the only other actively transmitting virulent β-coronavirus. MERS-CoV is the causative agent of Middle Eastern Respiratory Syndrome (MERS) and is more virulent but less infectious than SARS-CoV-2 and is phylogenetically different from SARS-CoV-2 (<90% amino acid sequence homology). Both viruses have a positive single-stranded RNA genome of ~30 kb that is polyadenylated that encodes four structural proteins (spike [S], membrane [M], envelope [E], and nucleocapsid [N]) that play similar roles within each virus. The two viruses diverge with respect to the receptor used for cell entry, their virulent accessory proteins and the specific function(s) of the 16 non-structural proteins (nsp1 to nsp16). Nsp's are produced by viral proteinase cleavage of two large polyproteins encoded by *ORF1a* and *ORF1b*. *ORF1* is closest to the 5′ end and is directly translated from genomic RNA upon entrance into host cells and a ribosome skipping mechanism divides it into *ORF1a* and *ORF1b* (Knipe & Howley, 2013). Whereas MERS-CoV encodes at least five accessory proteins (*ORF3*, *ORF4a*, *ORF4b*, *ORF5*, and *ORF8b*), SARS-CoV-2 encodes at least six (*ORF3a*, *ORF6*, *ORF7a*, *ORF7b*, *ORF8*, and *ORF10* [Wu et al, 2020]). All proteins not encoded by *ORF1a* or *ORF1b*, must be translated from subgenomic RNAs (sgRNAs) (Yount et al, 2003; Brian & Baric, 2005). SgRNAs are generated via a mechanism termed discontinuous extension that uses short sequences of varying length (usually 6–12 nucleotides [nts]) termed Transcription Regulatory Sequences (TRS's) spaced between genes to pair a 3′ portion of the negative viral strand to a complementary 5′ leader sequence of around 70 nts. This is followed by extension of the negative strand to the 5′ end of the positive strand, generating a short negative strand sgRNA intermediate. The RNA intermediary is then replicated to generate a positive strand sgRNA that encodes viral protein(s) (Sola et al, 2015).

Annotating viral transcriptomes is fundamental to understanding virus biology, which is a key aspect in combating viral transmission, replication, and pathogenesis. Prior coronavirus outbreaks, such as the Severe Acute Respiratory Syndrome (SARS) outbreak in 2003 and the MERS outbreak that began in 2012 (Peiris et al, 2003; Assiri et al, 2013), has increased research on these viruses as well as coronaviruses of zoonotic origin from which human coronaviruses are thought to originate. Comparing transcriptional variation of different coronaviruses may reveal mechanisms behind their distinct pathogenicity and infectivity, and potentially explain the molecular etiology behind how species barriers are crossed. Systematically annotating specific differences in the transcriptional profiles of virulent coronaviruses that are buried within numerous metatranscriptomic data sets may shed new light on viral transmissibility and virulence. However, even a simple systematic comparison of their in vitro transcriptional profiles is lacking.

[1]Shanghai Institute of Immunology, Shanghai Jiao Tong University School of Medicine, Shanghai, China   [2]College of Chemistry, Sichuan University, Chengdu, China   [3]Department of Medicine, University of Connecticut Health, Farmington, CT, USA

Correspondence: lei.chen@sjtu.edu.cn; dorsett@uchc.edu
*Lin Lyu, Ru Feng, and Mingnan Zhang contributed equally to this work.

For newly emerged SARS-CoV-2 virus, sequencing plays an essential role in diagnosis and monitoring of strain evolution (Wu et al, 2020; Zhou et al, 2020). However, in general, sequencing data sets for SARS-CoV-2 and MERS-CoV were limited to the description of both viral and host transcripts generated during infection of in vitro cell lines as well as model organisms. Analysis of viral transcriptomes originating from different viral strains in humans is overlooked as suitable analysis tools are lacking.

Sequence homology plays an essential role in the functional annotation of viral genes. However, sequence homology alone does not guarantee protein expression as rapidly mutating RNA viruses can harbor sequence alterations that result in novel or mutated ORFs that are not transcribed nor expressed. Therefore, direct profiling of viral RNAs is the key step toward understanding which viral products can actually be generated. For SARS-CoV-2, direct profiling of viral RNAs produced in a cultured cell line was recently conducted using Oxford nanopore technology and identified the existence of a canonical and non-canonical viral transcriptome (Kim et al, 2020; Taiaroa et al, 2020 Preprint). For SARS-CoV-2, proteomics study on the viral protein also exists (Davidson et al, 2020). However, this type of proteomics study using Mass spectrometry is generally not used to discover novel peptide. It also suffers from inability to recover short peptides. Both sequencing studies and the proteomics study used isolated virus strains to infect the VERO cell line isolated from kidney epithelial cells of the African green monkey that does not initiate an IFN response upon infection. Although these studies establish a basic characterization of virus transcription, each study only characterize viral gene expression of a single viral strain and is unable to determine if viral transcriptional responses are altered in response to host immune responses (e.g., IFN).

We developed a bioinformatics pipeline CORONATATOR (CORONAvirus annoTATOR) to quantify viral gene expression and identify bonafide sgRNAs in numerous publicly available metatranscriptomic data sets. Beyond outlining the variation in sgRNA profiles and their relative expression, our analysis identified novel sgRNAs for several different coronaviruses. It also revealed the presence of a core sgRNA repertoire that is shared between SARS and SARS-CoV-2 and one that is unique to MERS-CoV. A subset of novel sgRNAs for SARS-CoV-2 and MERS-CoV appear to be evolutionarily conserved in related coronaviruses found in bat and pangolin. Finally, we show that the transcription of specific sgRNAs differs significantly in vitro and in vivo as well as between different coronaviruses.

## Results

### CORONATATOR profiles viral sgRNAs via alignment breakpoint analysis

To systematically identify and compare coronavirus sgRNAs, we sought to identify publicly available coronavirus transcriptomic data sets. As of 2021/09/10, more than 3410427 viral genome sequences were submitted to the Global Initiative on Sharing All Influenza Data (GISAID) (Shu & McCauley, 2017). However, few data sets contain the raw sequencing reads. Using the search term "coronavirus" along with manual curation, we located raw reads in a

total of 19 bioprojects within the NCBI Short Read Archive that contain 588 samples for SARS-CoV-2 as well as related coronaviruses, such as SARS and MERS (Table 1). We also used an additional data set with a single sample that was recently published (Kim et al, 2020).

To profile the sgRNAs present within these data sets, we developed an informatics pipeline CORONATATOR. It was designed for the utilization of sequences produced by highly accurate next-generation sequencing technology that permits identification of TRS sequences from individual reads. Direct RNA Sequencing on the Oxford Nanopore platform can also be used to profile viral sgRNAs but is currently not supported by CORONATATOR because of the limited data availability as well as its restrictions in terms of sequencing accuracy and read length bias (see the Materials and Methods section).

Briefly, raw reads were first aligned to their respective viral references, that is, SARS-CoV-2 (GeneBank ID NC_045512.2), SARS-CoV (GeneBank ID NC_004718.3), MERS-CoV (GeneBank ID NC_019843.3), or reference for other species of coronaviruses (Table 1). Specific sgRNAs were inferred from alignment breakpoint analysis that identified reads that spanned the junctions between the 5′ leader sequence and more distal genomic sequence. In the process, the TRS sequences used by different sgRNAs were also identified. If the reference contains gene annotation, a link between the sgRNAs and the annotated genes were also established (Fig 1A and see the Materials and Methods section).

CORONATATOR was designed to profile all possible breakpoints. However, to obtain bonafide sgRNAs, we removed both rare breakpoints and breakpoints that were inconsistent across samples. A breakpoint can be viewed as consists of two separate genomic positions (Fig 1A). We also analyzed non-sgRNA breakpoints, for which the 5′ position does not encompass the leader TRS. Our data suggested that non-sgRNA breakpoints are very rare (usually below 0.05% of total sgRNA breakpoints) and inconsistent, as these breakpoints were never identified in more than a single study. We therefore focused on sgRNAs formed with the canonical 5′ leader and a 3′ body.

### Most predicted coronavirus ORFs can be validated by sgRNA analysis

Many ORFs are annotated for SARS-CoV, SARS-CoV-2, and MERS-CoV based on consensus sequence annotation and the existence of some are disputed by proteomics as well as sequencing studies (Davidson et al, 2020; Wu et al, 2020). Only after examination of a large number of data sets from multiple studies were we able to confidently assign commonly annotated ORFs into one of three categories (core, low support and no support) (Table 2). Core ORFs are defined as ORFs whose sgRNAs were consistently observed in most samples we profiled and having canonical TRS, they coincide with the eight well-known and conserved coronavirus genes/ORFs (Fig 1B). Other ORFs with any type of sgRNA support were put in the category of low support and these sgRNAs all came with degenerate TRS sequence. Whereas ORFs lacking any sgRNA support were put in the third category. They did not have proteomics support either.

Identifying bonafide sgRNAs requires multi-study and multi-sample analysis as unique artefactual sequences are often

**Table 1.  Bioprojects used in study.**

| Bioproject | Number of experiments | Number of runs | Virus sequenced | vivo/vitro |
|---|---|---|---|---|
| PRJEB13360 | 46 | 46 | HKU1; OC43; NL63 | In vivo |
| PRJNA233943 | 134 | 134 | SARS-COV; MERS-COV | In vitro |
| PRJNA233944 | 136 | 136 | MERS-COV; SARS-COV | In vitro |
| PRJNA238265 | 43 | 43 | MERS-COV | In vivo |
| PRJNA277369 | 5 | 6 | HKU3; HKU5; MERS-COV | In vitro |
| PRJNA279442 | 10 | 20 | MERS-COV; SARS-COV | In vitro |
| PRJNA545350 | 16 | 16 | MERS-COV | In vivo |
| PRJNA580021 | 27 | 27 | MERS-COV | In vitro |
| PRJNA601736 | 2 | 2 | SARS-COV-2 | In vivo |
| PRJNA603194 | 1 | 1 | SARS-COV-2 | In vivo |
| PRJNA605907 | 8 | 8 | SARS-COV-2 | In vivo |
| PRJNA605983 | 9 | 9 | SARS-COV-2 | In vivo |
| PRJNA606159 | 9 | 9 | Various Bat coronavirus | In vivo |
| PRJNA606165 | 1 | 1 | Bat coronavirus RaTG13 | In vivo |
| PRJNA606875 | 7 | 7 | Various Pangolin coronavirus | Both |
| PRJNA607174 | 7 | 7 | SARS-COV-2 | In vivo |
| PRJNA610428 | 14 | 14 | SARS-COV-2 | In vivo |
| PRJNA615032 | 110 | 329 | SARS-COV-2 | Both |
| PRJNA624792 | 2 | 2 | SARS-COV-2 | In vitro |
| Kim et al (2020)[a] | 1 | 1 | SARS-COV-2 | In vitro |

[a]This data set was obtained from Open Science Framework (OSF), and other datasets were obtained from NCBI Short Read Archive (SRA).

generated during sequence library preparation or sequencing (Peng et al, 2015; Lebrigand et al, 2020). In addition, many non-canonical sgRNAs found in low abundance may be random aberrant transcripts without dedicated function (Kim et al, 2020). Therefore, only sgRNAs that are present in multiple studies and data sets are true sgRNA candidates. To classify each viral gene we considered factors such as sgRNA relative abundance, TRS conservation and the potential for leaky ribosome scanning that can be affected by start codon hijacking (Schaecher et al, 2007).

For SARS-CoV-2, 34 samples were kept after removing those with <20 sgRNA reads. To identify robust and consistent sgRNAs that represent the "core" repertoire, which we assign to our first ORF category, we pooled all sgRNAs identified for a specific virus using a weighted average approach (see the Materials and Methods section) and noted their relative abundance. At a relative abundance of 0.5%, eight canonical breakpoints emerged corresponding to eight sgRNA species that harbor eight well-described ORFs for SARS-CoV-2: *S*, *E*, *M*, *N*, *ORF3a*, *ORF6*, *ORF7a*, and *ORF8* (Figs 1B and C and S1 and Tables S1 and S2). The sgRNA breakpoints for these ORFs are situated between 9 and 162 nt upstream of the start codon. *N* is the most abundant core sgRNA, representing 54% of the core sgRNAs identified in all samples. The *E* sgRNA is the least abundant at 1.5%, and the only core protein not identified in recent proteomics studies (Bojkova et al, 2020; Davidson et al, 2020). *ORF7a*, *M*, *ORF3a*, *S*, *ORF8*, and *ORF6* are present at 10.6%, 8.4%, 6.9%, 6.1%, 5.9%, and 2.7%, respectively. Together, these eight core sgRNAs account for 70–100% of the total

sgRNAs depending on sample type (e.g. in vivo versus in vitro), viral strain, and read coverage (Fig 1B and Table S2).

Beside their high relative abundance, these eight core sgRNAs are also defined by a shared canonical body TRS with a conserved core sequence of "ACGAAC," which is unique to this group of sgRNAs. Furthermore, the same eight core sgRNAs, as well the core TRS sequence, were shared by SARS (Fig S1). The seven core sgRNAs for MERS following (*S*, *E*, *M*, *N*, *ORF3*, *ORF4a*, and *ORF5*) (Fig 1C) also use this core sequence, with the exception of N that has a TRS which contains "ACGAA."

A second category of ORFs generally have their sgRNAs present at lower relative abundance and does not use the full conserved TRS sequence. This category includes *ORF7b* in SARS-CoV-2 and SARS-CoV, *ORF3b* in SARS and *ORF4b*, and *ORF8b* in MERS-CoV. For SARS-CoV-2, *E* has an average relative abundance of 1.5% which is the lowest amongst the core ones, whereas *ORF7b*'s is only 0.02%. This low abundance or low efficiency in sgRNAs formation may result from the use of non-canonical TRS's. This group of sgRNAs do not use the conserved core TRS sequence as core sgRNAs do, meaning the sequence homology they rely on for recombination is always shifted a few bases from the core and quite often they contain mismatches between leader and body TRS.

Other predicted ORFs fell into the third category with no sgRNA support, at least in the data set we examined. When factor in evidence beyond sgRNA support, this category can be further divided into two sub-categories. The first would be no sgRNA support

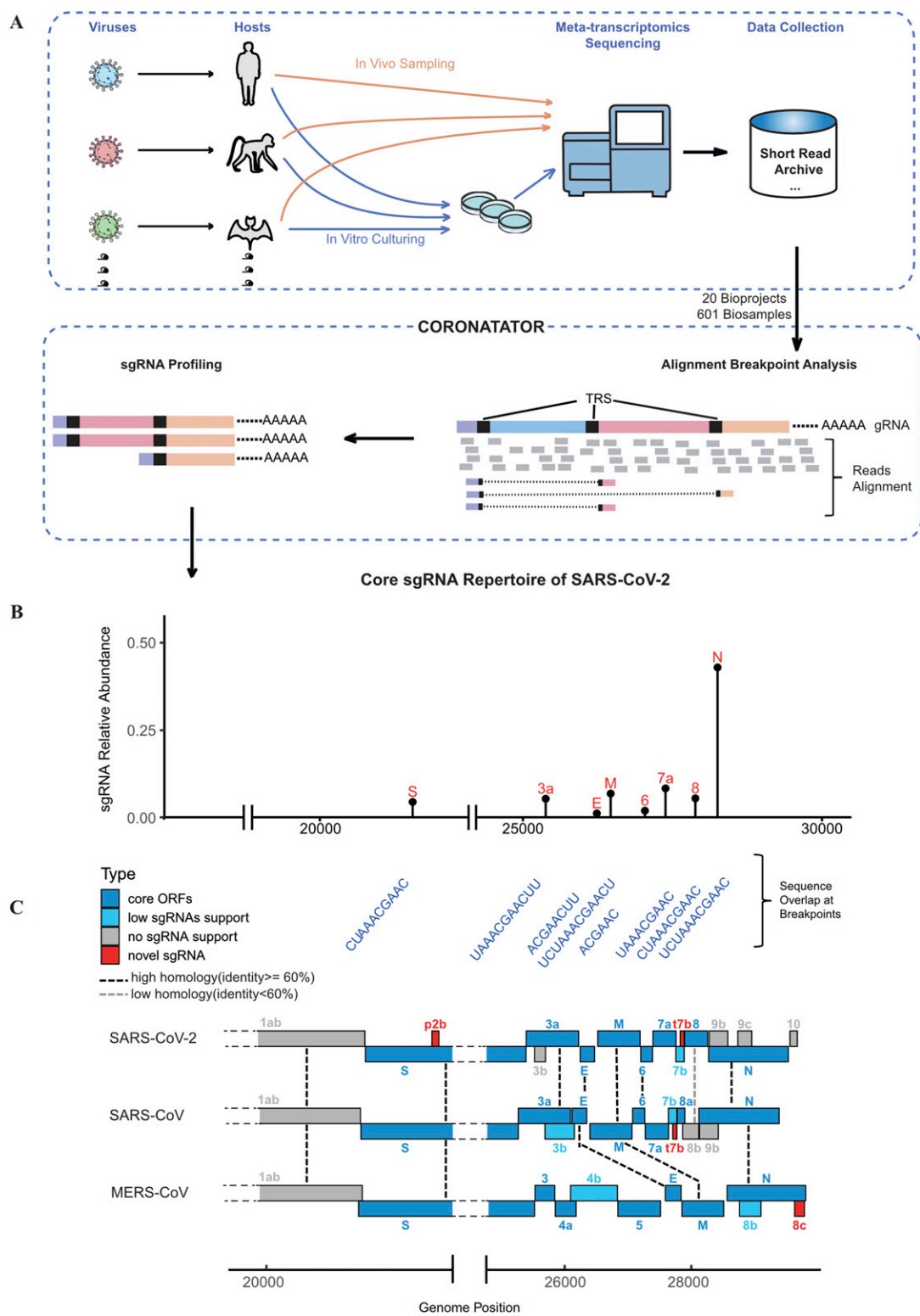

**Figure 1. Study overview and the subgenomic RNA (sgRNA) profile of SARS-CoV-2.**
**(A)** Study overview; Top panel: In vivo data sets used in this study came from different hosts infected by different coronaviruses. In vitro samples were obtained from virus infected cell lines. These samples were subjected to metatranscriptomic sequencing and reads were collected from Short Read Archive. Bottom panel: schema of mechanism behind CORONATATOR. **(B)** A set of SARS-CoV-2 sgRNAs that were consistently recovered from multiple data sets, y axis shows their relative proportion, they corresponded to the eight conserved ORFs, labelled in red. **(B, C)** Upper panel: the actual TRS sequence used by the sgRNAs shown in (B), note that they differ in length but contain a core conserved sequence. Lower panel: the conservations of the ORFs between SARS-CoV-2, SARS-CoV, and MERS-CoV, the ORFs were colored by their support level, novel ones were colored red. Dashed lines indicated conservation between species.

**Table 2. Support for commonly annotated SARS-CoV-2 ORFs.**

| ORF | Supported in present study | Supported by proteomics | Support level[a] |
|---|---|---|---|
| ORF1ab | − | + | Core ORF[b] |
| S | + | + | Core ORF |
| ORF3a | + | + | Core ORF |
| ORF3b | − | − | No support[c] |
| E | + | − | Core ORF[d] |
| M | + | + | Core ORF |
| ORF6 | + | + | Core ORF |
| ORF7a | + | + | Core ORF |
| ORF7b | + | − | Low support |
| ORF8 | + | + | Core ORF |
| N | + | + | Core ORF |
| ORF9b | − | − | No support |
| ORF9c | − | − | No support |
| ORF10 | − | − | No support |

[a]Core ORFs are defined as ORFs whose subgenomic RNAs (sgRNAs) were consistently observed in most samples we profiled and having canonical TRS, they coincide with the eight well-known and conserved coronavirus genes/ORFs. Other ORFs with any type of sgRNA support were put in the category of low support and these sgRNA all came with degenerate TRS sequence, whereas ORFs lacking any sgRNA support were put in the third category. They did not have proteomics support either.
[b]ORF1ab is directly translated from genomic RNA without forming sgRNAs.
[c]Though commonly annotated as a peptide generated via a leaky ribosomal scanning mechanism, ORF3b fell into the "no support" category as there are several start codons between ORF3a and ORF3b, there is no proteomic support and no unique ORF3b sgRNAs were identified.
[d]A well-known structural gene, not detected in proteomic study because of its short length.

but can potentially be translated, via a leaky ribosome scanning mechanism (Schaecher et al, 2007). *ORF9b* of SARS-CoV-2 falls into this sub-category. Indeed, multiple recent proteomics studies showed support for the ORF9b protein product in SARS-CoV-2 (Bojkova et al, 2020; Gordon et al, 2020). Its homolog in SARS-CoV, also named *ORF9b*, falls in the same category. Interestingly, *ORF7b* of SARS-CoV-2 and SARS-CoV were also supposed to be expressed in this fashion (Schaecher et al, 2007), and indeed the long stretch between start codons of *ORF7b* and preceding *ORF7a* (362 nt in SARS-CoV-2 and 365 nt in SARS) are void of additional start codons. Yet, our analysis suggested these gene products still form their own sgRNAs at low abundance.

The second sub-category contains the most suspicious ORFs, where sgRNA support cannot be found and intervening start codons between them and the closest sgRNA breakpoint would make their expression very unlikely. This category includes commonly annotated ORFs for SRARS-CoV-2 (*ORF3b*, *ORF9c*, and *ORF10*) and SARS-CoV (*ORF8b*). The several out of frame start codons between these ORFs and preceding ones, along with the absence of corresponding sgRNAs and their absence from proteomic studies (Bojkova et al, 2020; Gordon et al, 2020; Kim et al, 2020), strongly argues that these proteins are not generated.

For the commonly annotated ORFs of SARS-CoV-2, sgRNA support from this study and proteomics support from a previous study using Mass Spectrometry were summarized in Table 2. It is worth noting for all the ORFs detected by proteomics, corresponding sgRNA were also found. Whereas unlike mass spectrometry which has difficulty identifying small protein products, such as the envelope protein, our pipeline is able to predict the expression of small peptides even

at low concentrations through identification of low abundant bonafide sgRNAs.

### Identification of novel sgRNAs with non-canonical TRSs in SARS-CoV-2, MERS-CoV, and SARS-CoV

As mentioned before, during formation of the core sgRNA repertoire, a body TRS that contains a minimal core sequence will pair with the leader TRS. For each particular core sgRNA, the two TRS's used must be of the same length and sequence, although the length can vary between sgRNAs (Fig 1C). We found the average length of these canonical TRS's for SARS-CoV-2 was ~9.6 nts. Interestingly, the same core sequence is used in SARS, whereas MERS also uses a six nucleotide TRS with a different core sequence (Fig S1).

When we looked for sgRNAs that composed more than 0.2% of sgRNA transcripts, we identified three additional sgRNAs that were present in at least two separate samples and studies (Fig 2A). All three novel sgRNAs contained breakpoints that did not use canonical TRS sequences that are present in core sgRNAs. The three breakpoints support the discontinuous extension model of sgRNA formation, as the sequence from the body strand was found in the TRS sequences of the final transcript (Figs 2B and S2). On a separate note, sequence analysis of stranded RNA library preps identified the presence of negative strand sgRNAs, which were not described in the previous Nanopore sequencing articles (Davidson et al, 2020; Kim et al, 2020; Taiaroa et al, 2020 *Preprint*). As previously noted for artificial TRS's, analysis of these non-canonical breakpoint sequences revealed that TRS's without perfect complementarity may

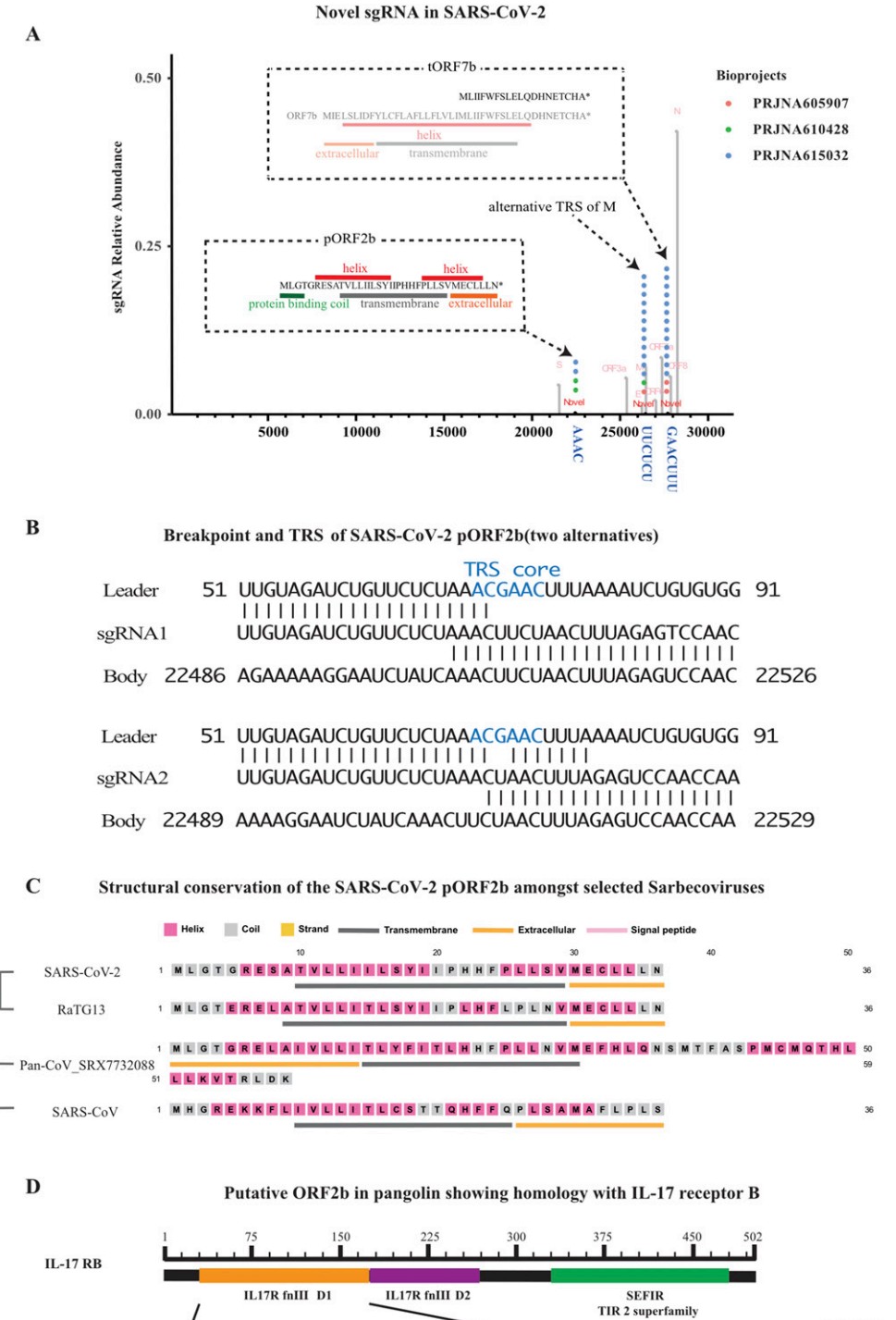

Figure 2. Novel subgenomic RNAs (sgRNAs) and prediction of ORF function for SARS-CoV-2. **(A)** Breakpoints plot for SARS-CoV-2 showing the three novel breakpoints at relative abundance cut-off of 0.1%, putative TRS sequences were shown below. Breakpoints for core sgRNAs were shown in grey as background. Peptides of novel ORFs, that is, putative *ORF2b* (*pORF2b*) and truncated *ORF7b* (*tORF7b*), were shown in inlets, secondary structures of these peptides were predicted and shown in different color. **(B)** Specially, a complete ORF7b peptide was shown in grey as a reference for the truncated one; (B) Sequence homology between leader TRS (top), sgRNA (middle), and body TRS (bottom) for novel sgRNAs. Homology to the canonical TRS was shown in blue. **(C)** Structural conservation of putative peptide translated from newly discovered *ORF2b*, shown as secondary structure prediction made by PSIPRED Workbench. A phylogenetic tree of examined coronavirus as determined by genomic sequences was also shown on the left. **(D)** Putative *ORF2b* in a pangolin CoV shows homology with cytokine receptor protein IL17RB's fibronectin III-like domain, a known IL17E ligand binding domain.

pair, and/or that large regions of complementarity around a core TRS between the body to itself, maybe used for the formation of sgRNAs (Fig 2B). Our analysis confirmed that TRS sequences can vary significantly between distantly related viruses and find that

canonical TRS sequences can be more than 30 nt in length in some coronaviruses (Fig S1).

The three novel TRS's generated three novel sgRNAs that we have termed putative *ORF2b* (*pORF2b*), alternative *M* (*aM*), and truncated

ORF7b (*tORF7b*). The longest novel sgRNA, *pORF2b*, is within the *S* gene and has two alternative TRS's positioned around 22501. Interestingly, it encodes a novel peptide that has a domain structure that is conserved in closely related coronaviruses, with at least one virus harboring and extended ORF (Fig 2B and C). The second novel breakpoint is located at 26494, 31 nt downstream of the canonical breakpoint for *M*. The sgRNA would support *M* expression, but with an alternative 5' UTR (Fig S2). The shortest of the three novel sgRNAs has its breakpoint positioned at 27761 and codes for a truncated version of *ORF7b* (*tORF7b*). The truncation removes the extracellular domain and 14 of the 24 amino acids that comprise the transmembrane domain (Figs 2A and S2). This sgRNA is expressed at relatively high levels both in vivo and in vitro and likely harbors novel functions (see the Discussion section below).

Translation of *pORF2b* results in a 36 amino acid peptide. It was predicated by PSIPRED (Buchan & Jones, 2019) to have a intracellular protein binding coil and two short $\alpha$-helixes that overlap a transmembrane domain, with the second $\alpha$ helix partially extracellular (Fig 2A and C). *pORF2b* was present in four samples in two separate studies. The highest expression of *pORF2b* was observed in a patient derived sample from Washington State in the United States (SRX7884411), where it accounted for a substantial 11.1% of the total sgRNAs. In a separate patient sample (SRX7884409) from the same bioproject, the novel ORF represented 1.1% of the sgRNAs identified (Table S2). The virus strains infecting these two patients differed by one nucleotide. Five other patient samples from the same study with different viral strains (Table S2) did not yield sgRNAs for *pORF2b*. The low breakpoint read numbers for these samples as well as viral strain may contribute to the variable detection of *pORF2b* in vivo. This indicates that the level of *pORF2b* transcripts maybe loosely correlated with viral strain and further demonstrates that samples within this bioproject are not cross contaminated with an artificial *pORF2b* sgRNA. SgRNA of *pORF2b* was also identified in a separate study (PRJNA615032), in two in vitro samples that used a different viral strain than any of those identified in the in vivo study (Table S2).

We searched for sequence conservation of *pORF2b* in other related Sarbecoviruses, including SARS-CoV, HKU3 (bat coronavirus), RaTG13 (a bat coronavirus proposed to be directly related to SARS-CoV-2), and a coronavirus infecting pangolin (SRX7732088) (Lam et al, 2020). A corresponding ORF was identified in all four viruses, with the highest level of homology found in RaTG13, with 91.89% nucleotide identity (Table S3 and Fig 2C). Interestingly, pORF2b and more so the pangolin version which has a C terminal extension, share high similarity with the ligand binding domain of human IL17RB (Fig 2D and see the "Discussion" section).

The third novel breakpoint was located at position 27761, within ORF7b, and encodes a truncated version of *ORF7b* (*tORF7b*). We identified this transcript in vivo and in vitro in two separate bioprojects that included more than one viral strain. This transcript was also recently identified in a VERO cell line infected by a single viral strain (Kim et al, 2020). Interestingly, a SARS-CoV homolog of this sgRNA was also present in several samples across two studies. This truncated version of ORF7b is missing the intracellular domain and more than half of its transmembrane domain, whereas retaining its hydrophilic extracellular domain (Fig S2). ORF7b is present in the SARS-CoV virion particle and is homologous to ORF7b

encoded by SARS-CoV-2 (Schaecher et al, 2007). The portion of ORF7b encoded by *tORF7b* is highly conserved in SARS (Table S3 and Fig S2).

We also obtained a significant amount of in vivo and in vitro sequence data sets for MERS-CoV, allowing us to identify abundant non-canonical sgRNAs (Fig S1). This novel sgRNA (putative *ORF8c* or *pORF8c*), is predicted to encode a ORF that translate into a novel 51 amino acid peptide. This novel sgRNA was identified in five separate studies, both in vivo and in vitro, ranging in abundance from 0.03% to 1.0% of total sgRNAs. PSIPRED suggest this novel peptide has a transmembrane domain connected to a cytoplasmic helix domain. We also looked for its conservation in other Merbecoviruses, including HKU4, HKU5, and an Erinaceus coronavirus. *pORF8c* could be found in all three with varying conservation (Fig S2 and Table S3). The cytoplasmic N terminal was the most conserved across Merbecovriuses and C terminal elongated versions were observed in HKU5 and Erinaceus (Fig S2).

To exhaust our search for novel sgRNAs, we lowered our threshold value to a relative abundance of 0.01%, whereas maintaining our other criteria. This analysis identified additional novel sgRNAs that appeared in more than one study for SARS-CoV-2, SARS-CoV, and MERS-CoV (Table S4). Additional sequencing and future experiments will determine the significance of *pORF2b*, *tORF7b*, and *aM* as well as the numerous other novel sgRNAs present at extremely low abundance.

To validate the experimental utility of our pipeline, we used an experimental data set that tested the effects of Gleevec and IFN-$\beta$ on host gene expression during treatment of MERS-CoV infection in vitro, from which we can observe significant expression change under different treatments (Fig S3).

### The relative abundance of Spike sgRNAs is elevated for SARS-CoV-2 in vivo

The relative abundance of a specific sgRNA to all sgRNAs in a particular viral sample is analogous to relative gene expression. There were large numbers of SARS-CoV-2 data sets examined in this study to enable the construction of a heat map. Information on sample origin and in vivo/in vitro status, as well as mutations identified in the different strains were plotted alongside the heat map to investigate their correlation with the pattern of gene expression (Fig 3 and see the discussion section below). Samples coming from the same institution did show tendency to cluster together, underscoring the effects of sample collection and experimental approaches on the result. Despite these batch effects, a strong pattern still emerged separating the in vivo samples from the in vitro ones.

In addition, while processing the data, we noticed two distinct patterns of read coverage along the SARS-CoV-2 reference genome that suggested that viral reads originated from two sources. Upon further examination, it was revealed the two sources were in vivo and in vitro derived samples (Fig S4). The former is composed of extracellular virion particles and infected host cells present in BALF (human) and nasal washes (Ferret) or lung homogenate (MERS), whereas the latter is composed of infected cells that are not subject to systemic or sometimes innate (e.g., VERO cells do not produce IFN) anti-viral responses. In vivo derived viral sequences obtained

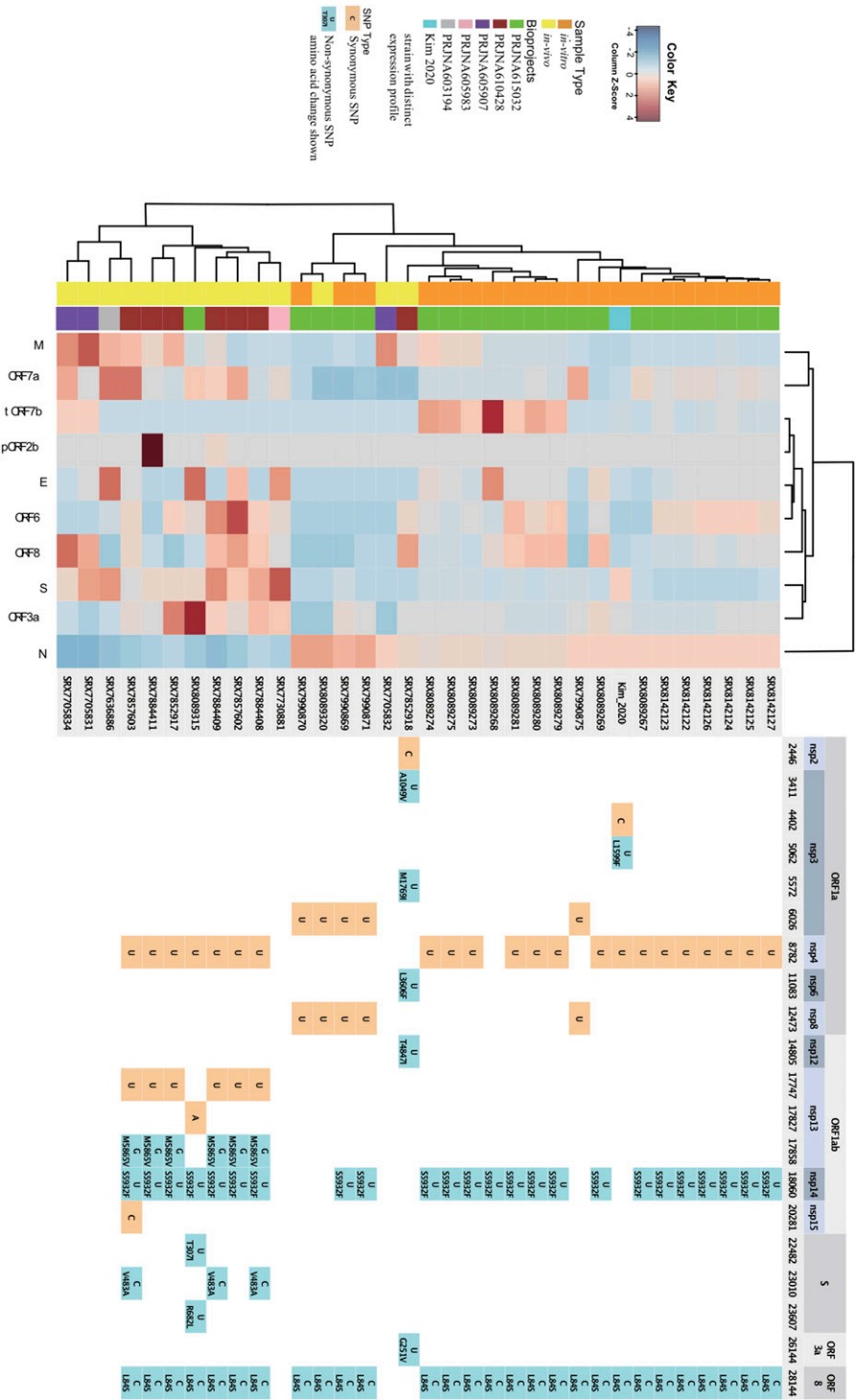

**Figure 3. Expression profile of SARS-CoV-2 from multiple data sets along with the underlying viruses' SNP annotation.**
Left panel shows the heat map of subgenomic RNA relative abundance for the multiple data sets that had enough subgenomic RNAs recovered, sample ID, usually an SRA ID, was also shown. Right panel shows where all the SNPs along the different viral genomes were located, synonymous and non-synonymous SNPs were colored differently. Virus strains from SRX7852918 and Kim et al (2020) had distinctive SNP pattern as well as characteristic expression profiles.

primarily from BALF for SARS-CoV-2 (primarily BALF) generally covered the entire viral reference length, with little bias towards the sgRNA containing 3' end. In contrast, highly elevated coverage at the 3' end of the viral genome was observed in the in vitro samples because of the formation of nested sgRNAs during viral transcription.

SARS-CoV-2 and MERS-CoV are the only coronaviruses still reported to infect humans and are present in both in vivo and in vitro derived metatranscriptomic data sets. We analyzed the relative abundance of sgRNAs generated in vivo and in vitro for both SARS-CoV-2 and MERS-CoV. When comparing the relative abundance of viral sgRNAs generated in vivo to those generated in vitro, it was

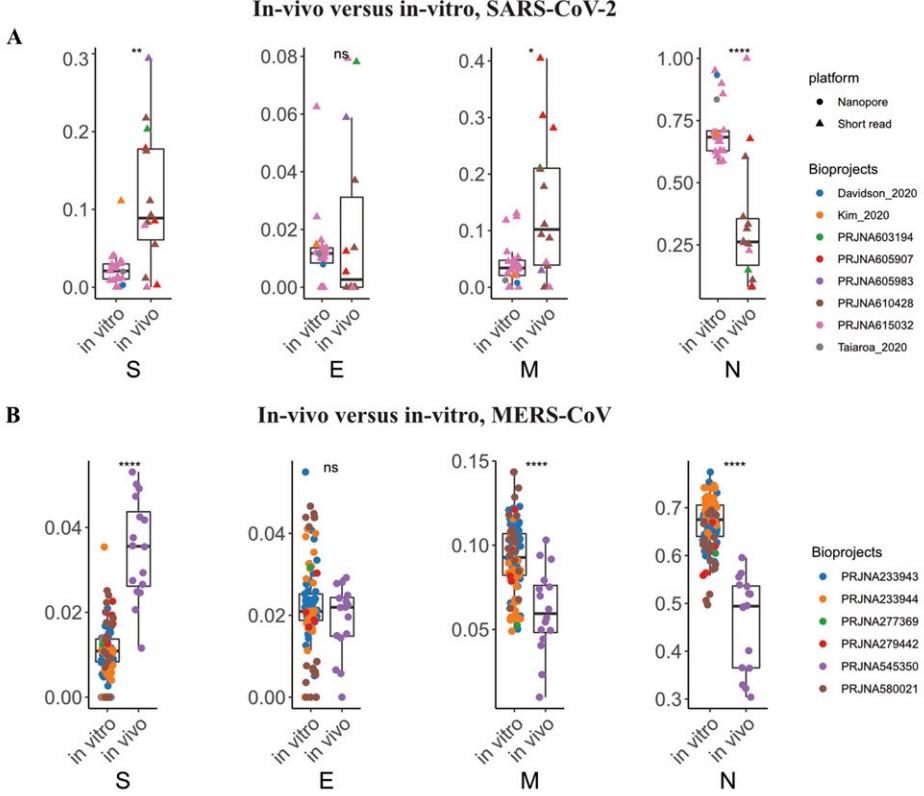

In-vivo versus in-vitro, SARS-CoV-2

In-vivo versus in-vitro, MERS-CoV

**Figure 4. Comparison of in vivo and in vitro subgenomic RNA expression.**
**(A, B)** subgenomic RNA expression profiles for SARS-CoV-2 and MERS-CoV, in vivo and in vitro data sets were grouped separately. It should be noted that two long read technology data sets were added to SARS-CoV-2 in vitro group, a math model was applied to adjust long read expression ratio to be comparable with short read data sets. Higher levels of *S* and *M* expression ratio and lower level *N* expression ratio were observed in in vivo sample versus in vitro sample in these two coronaviruses. **(C)** Left panel: phylogenetic tree of coronaviruses covered in this study; branch length indicates phylogenetic distance which was calculated as the ratio of nonidentical base positions to all base positions; the two major branches correspond to two genera. Right panel: expression ratio of Spike (*S*) genes in vivo and in vitro in different coronaviruses, each dot represents a sample, black bar indicate average expression level. **(A, B)** In (A, B), the *P*-values were calculated using Wilcoxon rank sum test and threshold are as follows: *P ≤ 0.05; **P ≤ 0.01; ***P ≤ 0.001; ****P ≤ 0.0001.

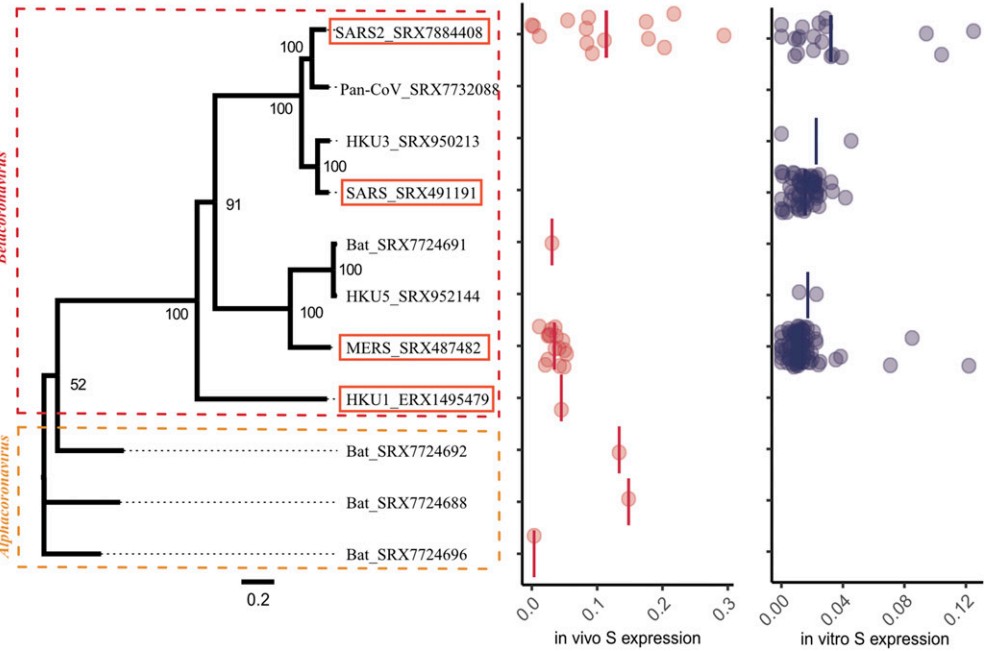

**C** Phylogenetic Tree of Involved Coronaviruses    Expression of S in Vivo Across Coronaviruses

evident that the ratio of *S* sgRNA to *N* sgRNA was significantly higher in vivo, especially for SARS-CoV-2 (0.04 in vitro versus 0.69 in vivo for SARS-CoV-2, *P*-value = 0.0012 with Wilcoxon rank sum test) (Figs 4A and B and S5). This difference may be due to the higher levels of

viral replication in vitro and/or altered viral gene expression to host defences in vivo, or has to do with different life phase the virus is in. As under in vitro conditions, the virus seemed to be in a more actively replicating phase, evidenced by the higher sgRNA to gRNA

reads ratio (Fig S4). Other examples of sgRNAs that are significantly differentially expressed in vitro and in vivo include the overall increase in the levels of accessory sgRNAs that act via multiple pathways to quell the immune responses to both SARS-CoV-2 and MERS-CoV (Fig S5; Canton et al, 2018).

To obtain a clearer perspective on how the relative abundance of SARS-CoV-2 sgRNAs compares to other coronaviruses in vivo and in vitro as well as determine if additional novel sgRNAs have been overlooked, CORONATATOR was used to analyze additional coronaviruses. This analysis included OC43, NL63, HKU1, as well as bat and pangolin viruses with high sequence homology to SARS-COV-2 (Lam et al, 2020; Zhou et al, 2020) (Figs 4C and S1 and Table 1). Some data sets did not yield enough breakpoint reads to be informative. For example, analysis of the bat virus RaTG13, with the highest homology to SARS-COV-2, yielded only one breakpoint read and was therefore omitted from Fig 4C.

Of the different coronaviruses profiled, SARS-COV-2 stands out as having the highest levels of S sgRNAs, especially in vivo (see the Discussion section below and Fig 4C). Our analysis indicates that this is independent of viral strain as it is present at high levels in different strains identified in vivo (Fig 3). The high levels of Spike protein may play a role in the viruses ability to cross the species barrier (see the Discussion section below) and its high rate of infectivity. In agreement, we noted that the relative levels of the Spike sgRNA is positively correlated with coronavirus infectivity. Viral infectivity and levels of S sgRNAs in vivo are as follows: SARS-COV-2 > HKU1 > MERS (Kissler et al, 2020). However, S protein levels alone are not sufficient to cause high levels of SARS-CoV-2 transmissibility, as factors such as Spike protein stability, receptor avidity (Wrobel et al, 2020) and virion stability (Aboubakr et al, 2020), also contribute to viral transmissibility. Also, we cautioned that this particular result was based on limited number of data sets, further sequencing and other studies are needed to validate this hypothesis.

### Mutations in the reverse transcription complex reverse the expression of *N* and *S* sgRNAs in vitro and in vivo

We also observed mutations in viral Reverse Transcription Complex (RTC) components that altered the expression profile of *S* to *N*. Specifically, the viral strain studied in Kim et al (2020) had one unique non-synonymous mutation in the RTC component nsp3, a papain protease that binds the N and M protein (Fig 3). The transcriptome generated in vitro for this viral strain showed a dramatic increase in the *S* to *N* ratio, mimicking the expression profile of viruses found in vivo (Figs 3 and S6). Interestingly, a viral strain identified in vivo (SRX7852918), had two non-synonymous mutations in nsp3, as well as nsp6 and nsp12 and had an in vitro like transcription profile, with a decreased *S* to *N* ratio (Fig S6). According to the discontinuous extension model, the RTC travels along the viral gRNA template and whenever an internal TRS (non-leader) homologous sequence is encountered, the RTC has a chance to switch template and sgRNA was made. This chance has to do with the sequence homology of the said TRS sequence and also could also be affected by intrinsic property of the RTC. Thus, coupled with the conserved order in how the core genes are situated along the viral genome, the observation presented here, suggested an elegant way of viral gene expression regulation.

## Discussion

The vast amount of sequence data generated for SARS-CoV-2 thus far has primarily been used for the typing and following of emerging viral strains. Although this is important, we felt such a focus could be an underutilization of a valuable information. By developing the CORONATATOR informatics pipeline, we took a step beyond the characterization of viral strains and described coronavirus viral sgRNA expression and uncovered novel and conserved sgRNAs. Functional prediction for some of these novel putative proteins is still ongoing. We tentatively show that a homolog of SARS-CoV-2 *pORF2b* in pangolin virus shares extensive similarity with human IL17RB's ligand binding domain. It is curious that a coronavirus may generate a peptide that could theoretically disrupt IL17B and IL17E (IL25) signalling as they are generally associated with promoting or inhibiting inflammatory responses in specific contexts. Future proteomic studies and/or ribosome sequencing studies will be required to verify the production of the protein products encoded by the novel sgRNAs identified here.

*ORF10*, a commonly annotated viral gene for SARS-CoV-2, is not supported by our analysis. It was also debated in recent studies (Davidson et al, 2020; Taiaroa et al, 2020 *Preprint*). The evidence described above indicated the potential pitfalls of conducting experiments on viral products from putative ORFs with no sgRNA or proteomic support. For example, a recent study that generated a synthetic version of the predicted truncated version of *ORF3b* in SARS-CoV-2 speculated that the putative truncated version in SARS-CoV-2 had a stronger anti-IFN activity than the SARS version (Konno et al, 2020).

The analysis presented here also implicates that different strains of coronaviruses express sgRNAs at different levels (Table S2 and Fig 3), especially for the non-core ones. These viral genes seemed to be dispensable, whereas capable of conferring specific advantages at certain conditions. Intriguingly, a previous study observed that a 45 nt deletion in SARS *ORF7b* that removes much of the transmembrane domain lost in tORF7b, attenuated the induction of IFN-β, provides a replicative advantage in vitro and in vivo as well as to cells pretreated with IFN-β (Pfefferle et al, 2009). Future research will reveal if this novel sgRNA encodes a novel virulent peptide that has function(s) antagonistic to IFN, while subverting the initiation of an IFN response.

The differences in environmental pressures that influence the requirement for these sgRNAs for viral replication, provide a general explanation for the striking variation in sgRNA levels in vivo versus in vitro. For example, the primary function of the S protein centers (higher in vivo) around host cell recognition and invasion, whereas the primary function of the N protein centers (higher in vitro) around the regulation of viral RNAs to promote viral replication (Molenkamp & Spaan, 1997; Fan et al, 2005; Liang et al, 2020). This suggests that molecular mechanisms, such as those that promote TRS readthrough for long sgRNAs, such as for *S*, may be responsive to the viral replication state and/or host signals (Wu et al, 2014.). Future electron microscopy studies on in vivo and in vitro virion particles will determine if Spike sgRNA abundance in SARS-CoV-2 correlates with spike protein levels on virion surfaces.

In the RTC complex, the observation that mutations in nsp3 occur in the two viruses with altered gene expression is thought provoking. Nsp3 is reported to bind TRS's, the 3′ end of the viral genome, the global viral RNA packaging signal as well as the N and M proteins (Hurst et al, 2013; Lei et al, 2018; Liang et al, 2020). In addition, phosphorylation of the N protein has been reported to alter its conformation to preferentially bind viral RNA and promote TRS readthrough during the generation of long sgRNAs (Chang et al, 2014; Wu et al, 2014). This observation tentatively implies that mutations within nsp3 affect the relative abundance of sgRNAs by acting in a global mechanism that influences overall viral structure and may act in concert with the mechanism described above for infectious bronchitis virus.

Our findings underscore that a true understanding of viral pathogenesis in terms of sgRNA expression can only come from thorough sequencing of patient samples in which the virus is under selective pressure. This begs for in-depth case examination, in which thorough sequencing and analysis is conducted for different stages of COVID-19 on a strain by strain basis. This would result in truly individualized patient care.

Although other zoonotic viruses may share extensive sequence similarity to SARS-CoV-2 at the gene or genomic level, similarity alone is not sufficient for the generation of pathogenic human viruses. Generally ignored during discussion of zoonotic viral origins, the specific expression level of viral genes, such as the Spike protein, are likely important for crossing the species barrier. For example, considering the vast number of un-sampled zoonotic viruses, it is likely Spike proteins capable of crossing the species barrier already exist, yet are not expressed at sufficiently high levels to enable sustainable inter-human transmission. However, low level Spike protein expression would allow sporadic transmission from bat to human, yet would not be sustainable as human to human transmission would be low because of low S protein expression as well sanitary environments that do not exist for bats. In agreement, it has been observed that people living in proximity to bat caves harbor virus specific antibody without ever experiencing severe disease (Wang et al, 2018).

Our analysis of the metatranscriptomic data sets identified numerous sources of RNA, such as host RNA as well as microbial RNA (although not optimally captured). In a time when it is unclear why some people succumb to SARS-CoV-2 infection, whereas others do not, these valuable sequences should not be wasted and could be made more useful if more clinical information is shared for these data sets. Most GISAID entries for SARS-CoV-2 have a metatranscriptomic data set that supports it. However, current GISAID entries that simply outline the viral genome sequence and strain far out-number the raw read entries we identified in Short Reads Archive (SRA). Sharing the raw read information will greatly help researchers study this virus and ultimately curb it.

# Materials and Methods

### Data collection

All sequencing data used were collected from NCBI SRA. Some nanopore data sets were downloaded from online repository described in their respective manuscripts (Kim et al, 2020). The bioprojects were located by searching with key words "coronavirus" and with manual curation, only metatranscriptomic data were kept Raw reads files were downloaded from SRA using wget with a customized script, SRAtoolkit were used to generate compressed fastq files from downloaded sra files. After initial sequence alignment using bwa with reference genome sequences of SARS-CoV, SARS-CoV-2 or MERS-CoV, samples with too few viral reads were filtered out. CORONATATOR only uses reads generated from second generation technologies (Illumina), nanopore data were used for comparison.

### CORONATATOR

CORONATATOR were a series of perl and bash scripts developed for profiling and analysis of RNA-seq data from coronavirus. It consists of three major steps, including preprocessing, breakpoint identification, sgRNA calling, and profiling, details below.

#### Preprocessing

BAM files were generated from sequence alignment with reference genomes of SARS-CoV, SARS-CoV-2, or MERS-CoV, for viruses from bat and pangolin, responsive genome assemblies were obtained from NCBI as references. Single nucleotide polymorphisms (SNPs) were called and filtered with bcftools (Li, 2011) and annotated with vcf-annotator (https://github.com/rpetit3/vcf-annotator). In addition, consensus genome sequences were also generated with filtered SNPs for further analysis.

#### Breakpoint identification

Breakpoints were identified from alignments with soft or hard clips, these alignments were all partial alignments largely caused by reads with recombination joints, which was generated by the mechanism through which coronavirus produce their sgRNA. In this step, a matrix of reads' information, breakpoint sites, CIGAR strings together with possible TRS sequences, was generated.

#### sgRNA calling and profiling

Typical sgRNAs were identified and defined by two breakpoint coordinates on a reference genome sequence, these sites were obtained by extracting breakpoints from partial alignments, that is, one from primary alignment and the other from supplementary alignment. To recognize possible TRS pattern, sequences between breakpoint pairs were extracted from previous generated consensus genome sequences. After that, corresponding genes of called sgRNAs were identified by manually comparing the distances between start codons of known viral genes and their breakpoints. Biosamples with more than 20 sgRNAs were used for further analysis, in these samples, sgRNAs were counted by genes and normalized by total sgRNA count to obtain a transcription profile matrix.

#### Novel ORF identification

Potential ORFs were predicted using Prodigal (Hyatt et al, 2010) with -s arguments to write all potential genes. An in-house python script

was also used to identify very short ORFs. Then for sgRNAs with multiple bioproject support, we calculated and sorted the distances between their breakpoints and all identified start codon sites. ORFs that start closest to upstream breakpoints were bookmarked and manually checked for verification.

### Sequence alignment and phylogenetic analysis

Consensus genome sequences of SARS-CoV-2, SARS-CoV, MERS-CoV, and biosamples from bat or pangolin or other human coronavirus with more than 20 sgRNAs were used for phylogenetic analysis. Multi-sequence alignment were performed with MAFFT (Katoh et al, 2002), Maximum likelihood consensus trees were constructed using IQ-TREE (Minh et al, 2020) with 1,000 bootstrap times.

### Converting nanopore sgRNA proportion to short reads'

Kim et al (2020) included both nanopore data and short read data. The ratios between the two were used to convert the other nanopore data sets to proportions comparable with others in this study.

### Plots and statistical analysis

Heat maps showing gene expression profile were produced using "heatmap.plus" package. SgRNA expression dot plots and boxplots were made with "ggplot2" package to compare difference between gene expression among different sample origin, $t$ test and Wilcoxon test were used for statistical analysis.

### Function annotation

Novel peptide sequences were aligned with EMBL online tool FASTA (https://www.ebi.ac.uk/Tools/sss/fasta/) against UniProtKB/Swiss-Prot database with default arguments. NCBI CD Blast online service was used to identify protein domains.

### Sequence conservation

To check for sequence conservation of putative peptides in related viral species, we generated a reference database containing all predicted ORFs from related viral genomes. DC MegaBlast (Dis-Continuous MegaBlast) was used to search for inter-species homologs. Arguments were set as follows: window_size 0, gapopen 0, gapextend 2, penalty −1, reward 1, and num_alignments 1. A group of homologous ORFs were then subjected to multiple sequence alignment using MAFFT. After that, CLUSTAO (Clustal Omega) was used to calculate an identity matrix for the multiple sequence alignment result. The same procedure was performed for both nucleotide and amino acid sequences.

## Data Availability

The code of CORONATATOR is available in the GitHub repository (https://github.com/15274972986/CORONATATOR).

All used sequencing data are accessible with accession number provided in Table 1.

## Supplementary Information

## Acknowledgements

We thank Dr. Qiming Liang of Shanghai Institute of Immunology for his insightful suggestion.

### Author Contributions

L Lyu: formal analysis, investigation, visualization, methodology, and writing—original draft, review, and editing.
R Feng: resources, investigation, and writing—review and editing.
M Zhang: investigation.
X Xie: visualization.
Y Liao: investigation.
Y Zhou: supervision.
X Guo: supervision.
B Su: supervision.
Y Dorsett: investigation, visualization, and writing—original draft, review, and editing.
L Chen: supervision, methodology, and writing—original draft, review, and editing.

### Conflict of Interest Statement

The authors declare that they have no conflict of interest.

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
