## [Reviewer comments · Life Science Alliance]

Life Science Alliance

Sub-genomic RNA profiling suggests novel mechanism in coronavirus gene regulation and host adaption

Lin Lyu, Ru Feng, Minnan Zhang, Xiaoqing Xie, Yinjing Liao, Yanjiao Zhou, Xiaokui Guo, Bing Su, Yair Dorsett, and Lei Chen
DOI: 10.26508/lsa.202101347

Corresponding author(s): Lei Chen, Shanghai Institute of Immunology, Shanghai Jiao Tong University School of Medicine and Yair Dorsett, Department of Medicine, University of Connecticut Health

Review Timeline:

Submission Date:	2021-12-22
Editorial Decision:	2022-02-09
Revision Received:	2022-03-17
Editorial Decision:	2022-04-04
Revision Received:	2022-04-11
Accepted:	2022-04-11

Transaction Report:

February 9, 2022

Re: Life Science Alliance manuscript #LSA-2021-01347-T

Prof. Lei Chen
Shanghai Institute of Immunology
Shanghai
China

Dear Dr. Chen,

Thank you for submitting your manuscript entitled "Large scale sub-genomic RNA profiling suggests novel mechanism in coronavirus gene regulation and host adaption" to Life Science Alliance. The manuscript was assessed by expert reviewers, whose comments are appended to this letter. We invite you to submit a revised manuscript addressing the Reviewer comments.

Thank you for this interesting contribution to Life Science Alliance. We are looking forward to receiving your revised manuscript.

Sincerely,

B. MANUSCRIPT ORGANIZATION AND FORMATTING:

Reviewer #1 (Comments to the Authors (Required)):

This manuscript reports a thorough bioinformatic study of available coronavirus RNA profiling data, including SARS-CoV2. The authors report the expression of subgenomic RNAs (sgRNAs) in SARS-CoV2, SARS-Co and MERS-Co, which are particularly enriched in Spike and Nucleocapsid sgRNAs relative to the bulk of coronavirus analyzed. Differences between data obtained from infected individuals (in vivo) compared to transcriptomic data obtained in vitro using infected cells (lacking IFN response) are also reported. Overall, the conclusions are supported by the results. In addition, the article is very well written and of broad interest.

Main concern

While the authors claim the importance of analyzing a large number of transcriptomic data using accurate bioinformatic methodology to better annotate the viral genome, they did not analyze the available proteome studies. This is important, as they infer from the transcriptomic data that certain sgRNA species could affect coronavirus infectivity and transmission based on the higher ratio of S and N (as pointed out in the Abstract and the Discussion). Despite the lower availability of proteome data (viral- and host-encoded proteins) authors should consider including these data on their bioinformatic analysis to complement the transcriptome analysis.

Reviewer #2 (Comments to the Authors (Required)):

Lyu et al, present an interesting pipeline for the identification and quantification of sgRNA from SARS-CoV-2 (meta)genomics and (meta)transcriptomics libraries. The manuscript is a wide collection of data but it does not tell a story. Experiments are not introduced properly (with a scientific rationale) and data are widely discussed in the results section and this makes the manuscript very hard to read and undermine its scientific value. Below, you can find suggestions to improve the readability of the manuscript and ensure its publication

-Overall it is important that the authors presents the bio-projects used in their analysis and for clarity, I would suggest that supplementary table 1 is included in the main manuscript as Table 1.

-line 127-129 comes out of nowhere. I would suggest to focus on the breakdown points and the sgRNA in the first paragraph.

-it is unclear how the authors assigned sgRNA according to core, low support, no support. Are these definitions and the abundance cutoff used in other studies? If not, can the authors provide a scientific rationale for the cutoff?

-It is confusing for the reader how the relative abundance is calculated. Why is it relative to the pooled sgRNAs instead of being relative to the genomicRNA or in the case of in vitro dataset a non-viral housekeeping gene, such as GAPDH or actin? Would the analysis and the abundance of selected sgRNA change in that case? Wouldn't this be more physiologically relevant and avoid the bias of the different origin of samples and methodology used to generate the data?

-line 211-216, 282-287, belong to discussion

-line 313. If this paragraph only shows supplementary data, it can be summarized in a sentence and included in the previous paragraph as a validation. In addition, in line 320-321, this statement is not clear. How is it possible that the decreased viral load does not affect but increase instead the abundance of viral genes/proteins? Please, clarify

-Can the difference observed in S versus N sgRNA and abundance in vitro versus in vivo could be due to the active replication "status" of the virus. In other words, it is possible that in the in vivo studies from patients BALB or animal organs, the virus is not actively replicating or its replication is abortive whereas in the in vitro studies, the virus is actively replicating (because the infection conditions are controlled). It is very hard to compare the datasets in terms of relative abundance cause viral load data are not provided and in general in vivo specimens are more heterogeneous. I do not think that this observation is important or conclusive. In addition line 348-363 provide a discussion of the data that does not belong to the result section. I would focus on the second part of the paragraph, i.e. line 377 and figure 4C. It is an interesting observation but I would tone down the

conclusions and avoid overclaims.

-line 404-414 belongs to discussion. Because no primary figures are described in this paragraph, the findings could be summarized in a sentence and included in the previous paragraph.

Minor changes:

-Line 94: what does "basic immune response" mean?

Epithelial cells will only produce type I and type III IFN, not IFN gamma (that is produced by immune cells)!

Line 206: between them

Line 339: What does "active virulent" mean? The authors maybe referred to actively replicating?

-line 379 typo: coronatator

Reviewer #1 (Comments to the Authors (Required)):

This manuscript reports a thorough bioinformatic study of available coronavirus RNA profiling data, including SARS-CoV2. The authors report the expression of subgenomic RNAs (sgRNAs) in SARS-CoV2, SARS-Co and MERS-Co, which are particularly enriched in Spike and Nucleocapsid sgRNAs relative to the bulk of coronavirus analyzed. Differences between data obtained from infected individuals (in vivo) compared to transcriptomic data obtained in vitro using infected cells (lacking IFN response) are also reported. Overall, the conclusions are supported by the results. In addition, the article is very well written and of broad interest.

Main concern

While the authors claim the importance of analyzing a large number of transcriptomic data using accurate bioinformatic methodology to better annotate the viral genome, they did not analyze the available proteome studies. This is important, as they infer from the transcriptomic data that certain sgRNA species could affect coronavirus infectivity and transmission based on the higher ratio of S and N (as pointed out in the Abstract and the Discussion). Despite the lower availability of proteome data (viral- and host-encoded proteins) authors should consider including these data on their bioinformatic analysis to complement the transcriptome analysis.

Response: We are very grateful for these comments and we agree that more in depth comparison with results from proteomics study is important. Accordingly, we highlighted importance of proteomics study by Davidson et al 2020 in the introduction part of the manuscript, lines 78 through 80. We also included an additional Table 2 showing for each individual ORF, the support from our transcriptomic study versus support from proteomics study. Based on this table we can see that the envelope protein, a well-known structure protein was not detected due to its short length by proteomics (well detected by sgRNA). The novel ORFs detected in this study were also short and found to be strain and condition specific. Thus, we think it's better to share our findings and let better equipped and interested parties carry out more targeted proteomics verification or discovery. Actually, the fact that all of the commonly annotated ORFs that can be detected by proteomics were also well supported by our sgRNA profiling gave confidence in our approach and novel ORF discoveries.

Reviewer #2 (Comments to the Authors (Required)):

Lyu et al, present an interesting pipeline for the identification and quantification of sgRNA from SARS-CoV-2 (meta)genomics and (meta)transcriptomics libraries. The manuscript is a wide collection of data but it does not tell a story. Experiments are not introduced properly (with a scientific rationale) and data are widely discussed in the results section and this makes the manuscript very hard to read and undermine its scientific value. Below, you can find suggestions to improve the readability of the manuscript and ensure its publication

-Overall it is important that the authors presents the bio-projects used in their analysis and for clarity, I would suggest that supplementary table 1 is included in the main manuscript as Table 1.

Response: Revised as suggested.

-line 127-129 comes out of nowhere. I would suggest to focus on the breakdown points and the sgRNA in the first paragraph.

Response: Thank you for providing comments on how to improve the manuscript. In the revised manuscript we rearranged the order of the results to improve clarity. Lines 127 to 129 in the original version were subsequently moved to Lines 277 to 284 in the revised manuscript.

-it is unclear how the authors assigned sgRNA according to core, low support, no support. Are these definitions and the abundance cutoff used in other studies? If not, can the authors provide a scientific rationale for the cutoff?

Response: Sorry for the confusion. In this part of the result, we were trying to validate the commonly annotated SARS2 ORFs by looking for sgRNA support in multiple datasets. So there was “support for ORFs” and then there was “sgRNA support”. We clarified this in the revised version. In terms of different levels of support, we were referring to the sgRNA support for commonly annotated ORFs, and we made an additional Table 2, providing evidence supporting the 14 commonly annotated ORFs, including sgRNA evidence, ones from proteomics study and notable exceptions. The rationale and methodological approaches can be found in line 131 to 136 in revised manuscript. To answer the question, these definitions were not used in other studies although a set of conserved and functionally important viral genes were known (Knipe et al 2013 and Davidson et al 2020). The cutoffs used were not set *a priori*, but rather for convenience, to accommodate the pattern that emerged when we examined large number of datasets from related coronaviruses: the group of ORFs known to be conserved in beta coronaviruses had their sgRNAs shown up in many datasets and usually reaching large proportion, and with conserved TRS sequences at the same time. These ORFs were termed the core ORFs and correspondingly their sgRNAs core sgRNAs. The other ORFs with any type of sgRNA support were put into a different category termed “low support” as their sgRNA support were sporadic and the TRS sequences used degenerate. The remaining ORFs with no sgRNA support fall into a third category. It’s worth noting that the ones in the no support category have no proteomics support either.

-It is confusing for the reader how the relative abundance is calculated. Why is it relative to the pooled sgRNAs instead of being relative to the genomicRNA or in the case of in vitro dataset a non-viral housekeeping gene, such as GAPDH or actin? Would the analysis and the abundance of selected sgRNA change in that case? Wouldn't this be more physiologically relevant and avoid the bias of the different origin of samples and methodology used to

generate the data?

Response: Very insightful questions. For the first question, we did not consider the house keeping genes from the host as samples were from different organisms and different tissues as well as various states of disease. For the second question, we previously compared using either the total number of pooled sgRNAs reads or the total number of viral reads (from gRNAs and sgRNAs) to calculate relative abundance and it did not change the interpretation of the results. Furthermore, we reasoned that the former is better because the ratio of the number of sgRNA reads to the number for gRNA is far higher in-vitro than in-vivo, reflective of the high rates of viral replication. We covered this in lines 285 through 294, and in lines 301 through 304. In addition, support for some sgRNAs came in only a few reads in several samples and dividing that small number by the number of reads for all viral reads would generate very small numbers. Lastly, using pooled sgRNA read counts is more straight-forward in presenting the relative expression of different sgRNAs as they add up to one.

-line 211-216, 282-287, belong to discussion

Response: Revised as suggested. The first section mentioned were moved to line 355-361 and the second section to line 364-369 in the revised manuscript.

-line 313. If this paragraph only shows supplementary data, it can be summarized in a sentence and included in the previous paragraph as a validation. In addition, in line 320-321, this statement is not clear. How is it possible that the decreased viral load does not affect but increase instead the abundance of viral genes/proteins? Please, clarify

Response: Good suggestion and revised as suggested, lines 273 to 275. In regard to your second question, we are not referring to the absolute value of the viral gene expression, but rather that there is a broadening, or increased dispersion of viral gene expression. We are trying to describe the observation that when viral load is high (meaning more viral reads detected amongst all reads), the ratio of the different sgRNAs is more stable across replicative experiments. This may be partly due to larger fluctuation/uncertainty when read counts were small. We dropped this description in the revised and more concise version.

-Can the difference observed in S versus N sgRNA and abundance in vitro versus in vivo could be due to the active replication "status" of the virus. In other words, it is possible that in the in vivo studies from patients BALB or animal organs, the virus is not actively replicating or its replication is abortive whereas in the in vitro studies, the virus is actively replicating (because the infection conditions are controlled). It is very hard to compare the datasets in terms of relative abundance cause viral load data are not provided and in general in vivo specimens are more heterogeneous. I do not think that this observation is important or conclusive. In addition line 348-363 provide a discussion of the data that does not belong to the result section. I would focus on the second part of the paragraph, i.e. line 377 and figure 4C. It is an interesting observation but I would tone down the conclusions and avoid

overclaims.

Response: We agree that the in-vitro and in-vivo samples capture different states of viral propagation, with in-vitro samples capturing actively replicating virus and in-vivo samples capturing a heterogeneous population with a significant portion of transcripts coming from virion particles. This assertion is supported by the coverage plots for SARS2 in-vitro samples in Supplementary Figure 1 that show a large 3' coverage bias, as would be expected for transcription of high levels of sgRNAs. The more uniform coverage for the in-vivo samples, is expected for virion particles as only a select few sgRNAs are packaged with the gRNA. These observations also held true for MERS (data not shown). Nevertheless, we consider it important to point out that in contrast to the simple inverse relationship between sgRNA abundance and length observed in-vitro, the spike sgRNA (the longest sgRNA) is selectively enriched relative to the nucleocapsid sgRNA (the shortest core sgRNA) in-vivo. This suggests that the virus can differentially regulate its gene expression to cope with different conditions. The in vivo samples were indeed heterogeneous yet this observation stands for all the virus (SARS2 and MERS) for which both in vivo in vitro samples existed and different studies were included. We gave a more balanced description in the revised manuscript (lines 281 through 284).

With regard to the second suggestion, line 348-363 of the original version were moved to the discussion (lines 370 to 379) in the revised manuscript. Also the conclusion based on Fig 4C were toned down as suggested (lines 325 to 327).

-line 404-414 belongs to discussion. Because no primary figures are described in this paragraph, the findings could be summarized in a sentence and included in the previous paragraph.

Response: Revised as suggested. Lines 380 through 387.

Minor changes:

-Line 94: what does "basic immune response" mean?

Epithelial cells will only produce type I and type III IFN, not IFN gamma (that is produced by immune cells)!

Response: Thank you for the correction, and we changed to the term IFN which is more general.

Line 206: between them

Response: Corrected.

Line 339: What does "active virulent" mean? The authors maybe referred to actively replicating?

Response: Sorry for the confusion, actually we meant the two did not die out as SARS-CoV

and were more lethal, unlike the other human coronaviruses causing common cold. We changed the term into “actively transmitting virulent”

-line 379 typo: coronatator

Response: Corrected.

April 4, 2022

RE: Life Science Alliance Manuscript #LSA-2021-01347-TR

Prof. Lei Chen
Shanghai Institute of Immunology, Shanghai Jiao Tong University School of Medicine
227 South Chongqing Road
Shanghai 200025
China

Dear Dr. Chen,

Thank you for submitting your revised manuscript entitled "Sub-genomic RNA profiling suggests novel mechanism in coronavirus gene regulation and host adaptation". We would be happy to publish your paper in Life Science Alliance pending final revisions necessary to meet our formatting guidelines.

- please address Reviewer 2's remaining comments
- please add an ORCID ID for secondary corresponding author; you should have received instructions on how to do so
- please add the Twitter handle of your host institute/organization as well as your own or/and one of the authors in our system
- please make sure that the author names in the system match with the author names in the manuscript
- please add a callout for Figure S2C in the main manuscript text
- please add Figure Legends for Figures S5 and S6

A. FINAL FILES:

B. MANUSCRIPT ORGANIZATION AND FORMATTING:

**Submission of a paper that does not conform to Life Science Alliance guidelines will delay the acceptance of your

manuscript.**

The license to publish form must be signed before your manuscript can be sent to production. A link to the electronic license to publish form will be sent to the corresponding author only. Please take a moment to check your funder requirements.

Sincerely,

Reviewer #1 (Comments to the Authors (Required)):

The authors have addressed the comments satisfactorily

Reviewer #2 (Comments to the Authors (Required)):

The authors addressed the comments in a satisfying manner. The revised version of the manuscript has remarkably improved. The last minor suggestion is to adjust the colors of the figure 4 and S4 and S5 because the individual dots are not very visible.

April 11, 2022

RE: Life Science Alliance Manuscript #LSA-2021-01347-TRR

Prof. Lei Chen
Shanghai Institute of Immunology, Shanghai Jiao Tong University School of Medicine
227 South Chongqing Road
Shanghai 200025
China

Dear Dr. Chen,

Thank you for submitting your Research Article entitled "Sub-genomic RNA profiling suggests novel mechanism in coronavirus gene regulation and host adaption". It is a pleasure to let you know that your manuscript is now accepted for publication in Life Science Alliance. Congratulations on this interesting work.

DISTRIBUTION OF MATERIALS:

Again, congratulations on a very nice paper. I hope you found the review process to be constructive and are pleased with how the manuscript was handled editorially. We look forward to future exciting submissions from your lab.

Sincerely,
